# Biodiversity from the Sky: Testing the Spectral Variation Hypothesis in the Brazilian Atlantic Forest

Tobias Baruc Moreira Pinon [1,2,*], Adriano Ribeiro de Mendonça [1], Gilson Fernandes da Silva [1], Emanuel Maretto Effgen [2], Nívea Maria Mafra Rodrigues [1], Milton Marques Fernandes [3], Jerônimo Boelsums Barreto Sansevero [4], Catherine Torres de Almeida [5], Henrique Machado Dias [1], Fabio Guimarães Gonçalves [6] and André Quintão de Almeida [7]

1   Department of Forestry and Wood Sciences, Federal University of Espírito Santo, Jerônimo Monteiro 29550-000, ES, Brazil; adriano.mendonca@ufes.br (A.R.d.M.); gilson.silva@pq.cnpq.br (G.F.d.S.); nivea.m.rodrigues@edu.ufes.br (N.M.M.R.); henrique.m.dias@ufes.br (H.M.D.)
2   Institute of Agricultural and Forestry Defense of Espírito Santo, Vitória 29010-935, ES, Brazil; emanuel.effgen@idaf.es.gov.br
3   Department of Forestry Engineering, Federal University of Sergipe, São Cristóvão 49100-000, SE, Brazil; miltonmf@academico.ufs.br
4   Department of Environmental Sciences, Federal Rural University of Rio de Janeiro, Seropédica 23890-000, RJ, Brazil; sansevero@ufrrj.br
5   Department of Fisheries Resources and Aquaculture, São Paulo State University (Júlio de Mesquita Filho), Registro 11900-000, SP, Brazil; catherine.almeida@unesp.br
6   Canopy Remote Sensing Solutions, Florianópolis 88032-005, SC, Brazil; fabio@canopyrss.tech
7   Department of Agricultural Engineering, Federal University of Sergipe, São Cristóvão 49100-000, SE, Brazil; andreqa@academico.ufs.br
*   Correspondence: tobiasbaruc@gmail.com

**Abstract:** Tropical forests have high species richness, being considered the most diverse and complex ecosystems in the world. Research on the variation and maintenance of biodiversity in these ecosystems is important for establishing conservation strategies. The main objective of this study was to test the Spectral Variation Hypothesis through associations between species diversity and richness measured in the field and hyperspectral data collected by a Remotely Piloted Aircraft (RPA) in areas with secondary tropical forest in the Brazilian Atlantic Forest biome. Specific objectives were to determine which dispersion measurements, standard deviation (SD) or coefficient of variation (CV), estimated for the n pixels occurring within each sampling unit, better explains species diversity; the effects of pixel size on the direction and intensity of this relationship; and the effects of shaded pixels within each sampling unit. The spectral variability hypothesis was confirmed for the Atlantic Forest biome, with $R^2$ of 0.83 for species richness and 0.76 and 0.69 for the Shannon and Simpson diversity indices, respectively, using 1.0 m illuminated pixels. The dispersion (CV and SD) of hyperspectral bands were most strongly correlated with taxonomic diversity and richness in the red-edge and near-infrared (NIR) regions of the electromagnetic spectrum. Pixel size affected $R^2$ values, which were higher for 1.0 m pixels (0.83) and lower for 10.0 m pixels (0.71). Additionally, illuminated pixels had higher $R^2$ values than those under shadow effects. The main dispersion variables selected as metrics for regression models were mean CV, CV for the 726.7 nm band, and SD for the 742.3 and 933.4 nm bands. Our results suggest that spectral diversity can serve as a proxy for species diversity in the Atlantic Forest. However, factors that can affect this relationship, such as taxonomic and spectral diversity metrics used, pixel size, and shadow effects in images, should be considered.

**Keywords:** unmanned aerial vehicle; hyperspectral RPA; red-edge vegetation index; secondary forest; Shannon; Simpson; Drone

## 1. Introduction

Tropical forests represent approximately 45% of the world's forest area and play a crucial role in maintaining biodiversity on Earth [1]. These forests have a high species richness and are considered the most diverse and complex ecosystems globally [2]. Species diversity refers to the variety of living organisms that coexist within an ecosystem. Several indices quantify ecosystem diversity, enabling comparisons between different types of vegetation and habitats [3]. The most prominent indices include the Shannon diversity index, the Simpson dominance index, the maximum diversity [4], and the Pielou equitability index [3].

Species diversity is affected by climate change, deforestation, pollution, and forest fires, among other factors, interfering with the evolution of species. The increasing frequency and intensity of extreme events of climate change complicate the measurement and monitoring of some species, especially in regions with tropical forests. This is because information collected in situ is typically used to estimate diversity indices, such as the occurrence of a species (density-based indices) and/or its structural parameters (coverage-based indices), such as its height and basal area. These field collections are time-consuming, costly, and cover small areas, hindering large-scale and short-interval analyses.

Thus, estimating and tracking the evolution of species diversity and richness across large areas of tropical forests poses significant challenges, which makes it necessary to find extrapolation methods, i.e., make inferences based on samples from the area/ecosystem of interest [5]. In this context, remote sensing is a promising tool to improve this extrapolation, being an indirect method to evaluate and estimate species richness and diversity patterns [6,7]. According to [6], ecosystem heterogeneity can be assessed through remote sensing data by classifying spectral information in habitats. However, caution is required, as rigorous data validation is necessary, and results cannot be extrapolated to regions with different habitats [8,9].

Therefore, models were developed by [10] to estimate species diversity using remote sensing imagery, introducing the Spectral Variation Hypothesis (SVH), which became a quantitative tool for improving species lists [6]. The SVH suggests that species richness is related to variations in spectral characteristics of images, typically measured using the standard deviation (SD) and coefficient of variation (CV) [6,10,11]. Examples include dispersion measures based on vegetation indices (VIs), reflectance bands, or one or more principal components. Mapping this variation can generate vegetation cover maps and, if the hypothesis that areas with greater spectral variability have greater botanical richness is confirmed, the SVH could describe diversity and richness at large scales.

In recent years, the SVH has been tested in various contexts and with a wide variety of remote sensing data (Table 1). But despite the potential of remote sensing data in biodiversity monitoring, the proposed relationship between spectral diversity and species diversity remains under debate, as some studies have identified inconsistencies [8,12]. Reference [13] argued that the SVH has been insufficiently tested due to observational limitations, and the metrics, methods, and sensors that could provide more reliable estimates of plant biodiversity remain unclear. The Normalized Difference Vegetation Index (NDVI) [14] is one of the most widely used index in spectral diversity studies [9,12,15,16] due to its sensitivity to vegetation greenness and shadow effects from soil and canopy [17,18]. However, is the NDVI the most appropriate VI? Does it work for all types of forests and habitats? And is it the best VI for tropical forests?

Unlike multispectral sensors, which are commonly used in SVH studies and may undersample the available information in the reflectance spectrum due to a limited number of spectral bands with widths up to hundreds of nanometers, hyperspectral sensors are narrowband sensors capable of acquiring nearly continuous reflectance spectra in many narrow bands for each pixel [19]. This capability provides hyperspectral remote sensing with a distinct advantage, as it enables the capture of detailed biochemical and biophysical information about plants, such as chlorophyll and water content, which allows for more precise vegetation analysis [20,21]. Furthermore, these sensors possess a large number of

bands across the electromagnetic spectrum (more than 100), facilitating the estimation and testing of numerous vegetation indices (VIs) in SVH studies. However, few studies have been developed with hyperspectral sensors to test the SVH hypothesis [22,23] due to its high cost, low availability, complexity of the data, and the possibility of being affected by atmospheric variations [20,24,25], showing that this type of sensor should be further explored. Reference [12] highlighted the need for further research involving this subject because, although empirical studies have validated the use of spectral diversity to estimate plant species diversity [11,26,27], others have criticized it as unstable and unreliable in all contexts [9,28].

**Table 1.** Key variables found in remote sensing studies that tested the SVH.

| Ecosystems | Diversity Type | Platforms | Vector and Sensors | Heterogeneity Index | Response Variable | Spatial Resolution | Temporal Resolution | Associated Metrics and Types of Models | Reference |
|---|---|---|---|---|---|---|---|---|---|
| Forest, grassland, mixed types, wetland, coastal, savanna, agricultural, agro-forest, and others | Alpha diversity, Beta diversity, and Gamma diversity | Satellite, RPA, airplane, and field | Multispectral, hyperspectral, panchromatic, multisensory, and LiDAR | Coefficient of variation, Rao's Q index, standard deviation, mean distance from centroid, Shannon's H index, and convex hull/volume | Species richness, Shannon, Simpson, Phylogenetic diversity index, Native species/family richness, Species abundance, Others | 3 cm to 500 m | Mono-temporal, time-series, and multi-temporal | R/R² linear model, multiple regression, machine learning, (Random Forest, SVM, etc.), PCA (Principal Component Analysis), ANOVA, and Kriging | [9,29,30] |

Spectral variability is derived from information present in pixels; therefore, the spatial resolution of images should be consistent with ecological assumptions to make accurate biodiversity inferences [31]. High-spatial-resolution hyperspectral images collected by Remotely Piloted Aircraft (RPAs) have great potential for testing the SVH [32]. Reference [33] evaluated the diversity of a tropical forest in Panama using hyperspectral data collected with an RPA and found that VIs successfully captured forest variability (*r* = 0.9).

Few studies have tested the SVH in secondary tropical forests in Brazil, and even fewer have used hyperspectral images from RPAs. According to [29,30], most SVH studies have been conducted in subtropical forests, followed by tropical and temperate forests. Moreover, most studies have been predominantly in Europe (Italy and Germany), the USA, and Asia (China and India), and only two have been conducted in Brazil [34,35], both focused on the Amazon biome. Although there are studies assessing forest biodiversity in the Atlantic Forest based on spectral diversity [32], there are no studies directly testing the SVH in this biome. The Atlantic Forest is a Brazilian biome known for its biodiversity richness [36] and high occurrence of endemic species [37–39]. Thus, this biome was chosen to test the hypothesis that species diversity correlates with spectral diversity.

This study is innovative in employing high-resolution RPA-based hyperspectral sensors to investigate how various spectral metrics reflect taxonomic diversity in the Atlantic Forest, one of the most biodiverse ecosystems on the planet. The research introduces a novel methodology for evaluating the Spectral Variation Hypothesis (SVH) at a local scale within this ecologically significant and threatened biome. By testing multiple spectral metrics and their correlation with species diversity indices, the study establishes an initial methodological framework that can guide future research in other tropical ecosystems. Furthermore, our study also innovates by testing the effects of pixel size and shadow on the relationship between spectral and taxonomic diversity. The limited use of sensors, spectral platforms, and data sources in the Atlantic Forest [40] has restricted the understanding of the complexities of the SVH and its applicability as a biodiversity monitoring tool. There-

fore, this work aims to fill this gap, positioning itself as a key reference for future studies on spectral and taxonomic diversity in the Atlantic Forest.

Our main objective was to test the Spectral Variation Hypothesis (SVH) using hyperspectral data (spectral bands and VIs) collected by an RPA to assess the relationship between spectral diversity and species diversity in secondary tropical forests of the Atlantic Forest biome. The specific objectives were to determine which dispersion measurements (SD or CV) estimated for the n pixels occurring within each sampling unit better explain species diversity; the effects of pixel size on the direction and intensity of this relationship; and the effects of shaded pixels within each sampling unit.

## 2. Materials and Methods

### 2.1. Study Area

The study was conducted in four Atlantic Forest remnants in southern Espírito Santo state, Brazil (Figure 1). These forest remnants were at different successional stages, classified according to Resolution No. 29/1994 of the Brazilian National Environment Council (Conama), using quantitative and qualitative variables described in the resolution to classify the vegetation [41] (Supplementary Materials). The region's phytophysiognomy is composed of semideciduous seasonal forests within the Atlantic Forest biome [42]. Forest areas at the early regeneration stage were approximately 10 to 25 years old. Intermediate-stage forest areas were approximately 45 to 70 years old. Forest areas at advanced regeneration stage had no evidence of large-scale impacts for at least 100 years. The disturbance history of the areas and the age of each successional stage were estimated by analyzing satellite imagery (CNES/Airbus and Maxar Technologies) using Google Earth Pro (version 7.3) software for private areas (Figure 1A–C) and the Pacotuba National Forest Management Plan [43] for the conservation unit (Figure 1D).

The region's climate is Aw, tropical, with a dry season in the winter, according to the Köppen classification [44]. The average temperature is 23 °C and the average annual rainfall is approximately 1400 mm. The terrain varies from flat to rugged, with altitudes ranging from 95 to 245 m. The predominant soil in the region was classified as Typic Hapludox (Latossolo Vermelho Amarelo) [45].

### 2.2. Field Data Sampling

The forest inventory was conducted between 25 July and 25 September 2022. Twenty plots with 30 × 30 m each were randomly chosen, six in overgrown pastures, six at the early successional stage, four at intermediate, and four at advanced successional stages. The plots were oriented in a north–south direction, maintaining a minimum distance of 25 m from the forest fragment edges. The four vertices were marked and georeferenced with sub-meter accuracy, estimated by a GNSS RTK GEOMAX ZENITH-16 G system (GeoMax, Widnau, Switzerland). The mean RMSE value for X and Y was 0.73 m.

Diameters at 1.3 m above ground level (D) were measured using a tape ruler, and total heights (H) were measured using a graduated ruler in living trunks of trees with D equal to or greater than 5 cm. A Suunto PM-5/360PC hypsometer (Suunto, Vantaa, Finland) was used for the largest trees. A total of 225 morphospecies from 1932 trees were sampled in the forest inventory.

Botanical material was collected and taken to the Herbário Capixaba (CAP) at the Federal University of Espírito Santo (UFES) for species identification. The five species with the highest importance values in the study areas were *Anadenanthera peregrina* (L.) Speg., *Tabernaemontana hystrix* Steud., *Actinostemon verticillatus* (Klotzsch) Baill., *Astronium concinnum* Schott, and *Ramisia brasiliensis* Glaz.

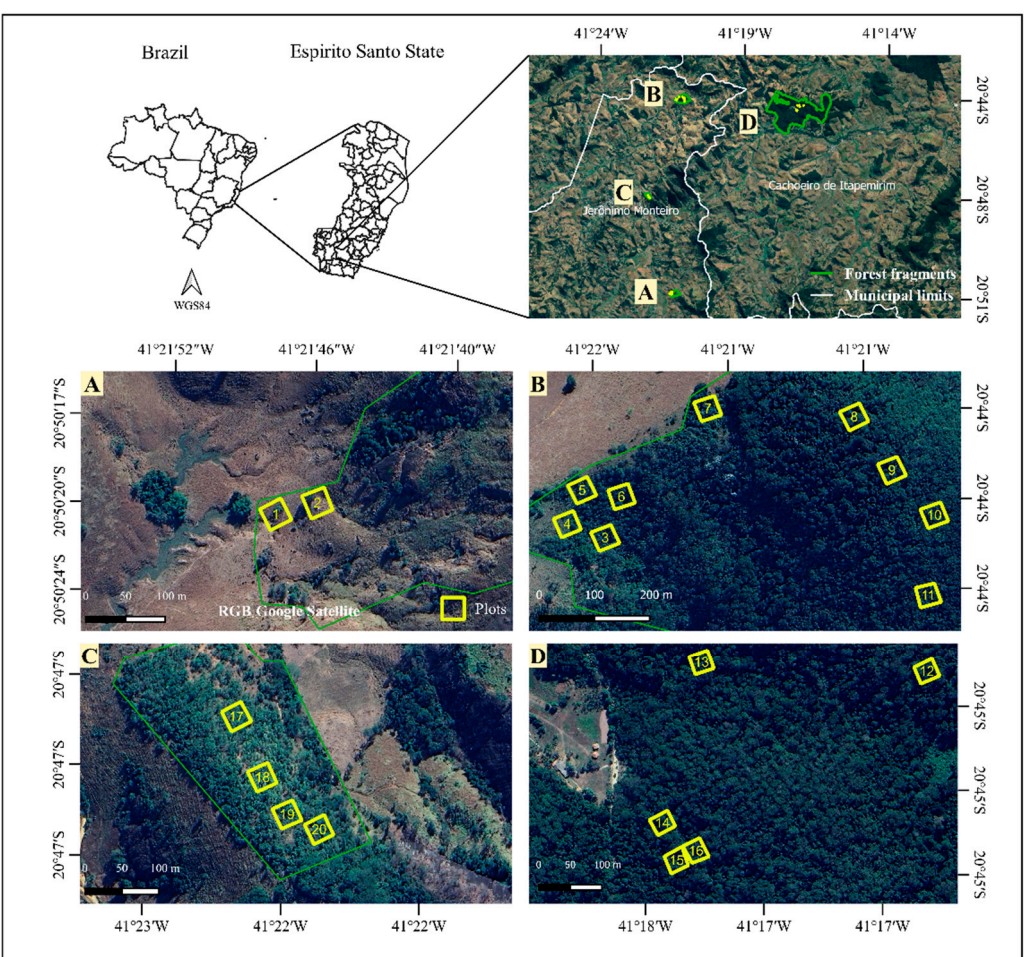

**Figure 1.** Location and layout of plots in the studied Atlantic Forest areas. Forests at the early successional stage: (**A**) plots 1 to 7 and (**B**) 17 to 20 (10–25 years); forests at the intermediate successional stage: (**C**) plots 8 to 11 (45–70 years); (**D**) forests at the advanced successional stage: plots 12 to 16 (more than 100 years old).

From the botanical and coverage (basal area) data obtained in the field, we calculated three biodiversity indicators: the Shannon and Simpson diversity indices and species richness. Species richness (S) was calculated by considering the total number of species within a plot. The Shannon (H′) and Simpson (C) diversity indices [3] were calculated with the following expressions:

$$H' = \frac{BA.ln(N) - \sum_{i=1}^{S} ba.ln(ba)}{BA} \tag{1}$$

where $ba_i$ = coverage of the *i*-th species (basal area); *S* = total number of species sampled; *BA* = coverage (basal area) of the plot; and *ln* = natural logarithm.

$$C = 1 - \frac{\sum_{i=1}^{S} ba_i(ba - 1)}{BA(BA - 1)} \tag{2}$$

where $ba_i$ = coverage of the *i*-th species (basal area); *S* = total number of species sampled; and *BA* = coverage (basal area) of the plot.

Descriptive statistics for total height, basal area, species richness, and species diversity indices can be found in Table 2.

**Table 2.** Statistics of total height of trees (H), basal area (BA), species richness, and Shannon and Simpson diversity indices for the study plots (n = 20).

| Statistics | H (m) | BA (m$^2$ ha$^{-1}$) | Richness | Shannon | Simpson |
|---|---|---|---|---|---|
| Mean | 8.31 | 16.70 | 20.20 | 1.37 | 0.56 |
| Standard deviation | 2.53 | 14.63 | 14.61 | 0.58 | 0.20 |
| Coefficient of variation | 0.30 | 0.88 | 0.72 | 0.43 | 0.36 |
| Minimum | 4.64 | 0.62 | 3.00 | 0.20 | 0.08 |
| Maximum | 12.21 | 46.07 | 58.00 | 2.26 | 0.77 |
| Median | 8.41 | 12.42 | 15.00 | 1.44 | 0.64 |

### 2.3. Remote Sensing Data Collection

Four aerial surveys were conducted in April 2023 using a multirotor RPA platform, model DJI Matrice 600 PRO (DJI, Shenzhen, China). The operations were performed within Visual Line of Sight (VLOS) and close to the local time of 12:00h. The flights were conducted at a maximum altitude of 120 m, in compliance with ICA 100-40 [46], with a longitudinal overlap of 10% and a lateral overlap of 40%. Weather conditions and visibility were good, and wind speeds were below 10 m s$^{-1}$ [47].

Hyperspectral data were collected using a Nano-Hyperspec sensor (Headwall, Bolton, MA, USA). This sensor captures wavelengths between 397.8 nm and 1002.3 nm, within 273 bands of 2 nm wide. Hyperspectral data were pre-processed, in which raw data cubes were converted into radiance (W m$^{-2}$ str$^{-1}$ nm$^{-1}$) using the HyperSpec III software [48]. Radiometric calibration and geometric correction were conducted using the same software. Radiance cubes were converted into surface reflectance using the field calibration method (calibration panel with known diffuse reflectance), as indicated in the manufacturer's manual [48]. Reflectance was calculated as the ratio of radiance to the mean radiance of calibration panel pixels. Solar incidence remained stable throughout the flight. Orthorectification of the surface reflectance cubes was performed using a Shuttle Radar Topography Mission (SRTM) digital elevation model with a 30 m spatial resolution. The orthorectification process began by calculating input geometry files (IGM) for each flight line, containing geographical coordinate information (datum WGS 84) for each original pixel of the raw image. The original pixel size of 0.11 m was resampled to 1.0, 5.0, and 10.0 m to improve the signal-to-noise ratio and evaluate the effects of pixel size, using the terra package [49] in the R programming environment [50].

### 2.4. Data Analysis

2.4.1. Hyperspectral Metrics

In addition to the reflectance of the 273 spectral bands, 30 vegetation indices (VIs) were estimated (Table 3).

**Table 3.** Vegetation indices and their respective equations and references. "ρ" indicates the reflectance of a hyperspectral band followed by its central wavelength in nanometers.

| Code | Index | Equation | Reference |
|---|---|---|---|
| C1 | Chlorophyll Index 1 | $(\rho850 - \rho710)/(\rho850 + \rho680)$ | [51] |
| C2 | Chlorophyll Index 2 | $\rho750/\rho700$ | [51] |
| AC1I | Anthocyanin Content Index 1 | $sum(\rho600/\rho700)/sum(\rho500/\rho600)$ | [52] |
| AC2I | Anthocyanin Content Index 2 | $\rho650/\rho550$ | [52] |
| PSI | Plant Stress Index | $\rho695/\rho760$ | [53] |
| SL | Slope of Red Edge | $(\rho740 - \rho690)/(N740 - 690)$ | [54] |
| NDVI | Normalized Difference Vegetation Index | $(\rho830 - \rho674)/(\rho830 + \rho674)$ | [14] |
| PRI | Photochemical Reflectance Index | $(\rho529 - \rho570)/(\rho529 + \rho570)$ | [55] |
| MEAN | Mean Reflectance Between 690 and 740 nm | $\Sigma i = 690$ to $740$ $Pi/N$ | [54] |
| MEDIAN | Median Reflectance Between 690 and 740 nm | $median \; \Sigma i = 690$ to $740$ $Pi$ | [54] |

**Table 3.** *Cont.*

| Code | Index | Equation | Reference |
|---|---|---|---|
| RVSI | Red-Edge Vegetation Stress Index | $(\rho714 + \rho752)/2 - \rho733$ | [54] |
| R1 | Ratio Vegetation Stress Index | $\rho694/\rho760$ | [53] |
| R2 | Ratio Vegetation Stress Index | $\rho600/\rho700$ | [53] |
| R3 | Ratio Vegetation Stress Index | $\rho710/\rho760$ | [53] |
| PSSR1 | Pigment Specific Simple Ratio 1 | $\rho800/\rho680$ | [56] |
| PSSR2 | Pigment Specific Simple Ratio 2 | $\rho800/\rho550$ | [56] |
| WBI | Water Band Index | $\rho970/\rho900$ | [57] |
| VARI | Vegetation Atmospherically Resistant Index | $(\rho557 - \rho643)/(\rho557 + \rho643 - \rho465)$ | [58] |
| SR | Simple Ratio | $\rho800/\rho680$ | [59] |
| NDVI2 | Normalized Difference Vegetation Index 2 | $(\rho800 - \rho670)/(\rho800 + \rho670)$ | [14] |
| EVI | Enhanced Vegetation Index | $2.5 \times (\rho897 - \rho673)/(\rho897 + 6 \times \rho673 - 7.5 \times \rho474 + 1)$ | [60] |
| SIPI | Structurally Insensitive Pigment Index | $(\rho800 - \rho445)/(\rho800 + \rho680)$ | [61] |
| CARI | Chlorophyll Absorption in Reflectance Index | $(\rho700 - \rho670) - 0.2 \times (\rho700 - \rho550)$ | [62] |
| CI.rededge | Chlorophyll Red Edge | $\rho851/\rho710$ | [63] |
| CI.green | Chlorophyll Green | $\rho730/\rho531-1$ | [63] |
| mARI | Modified Anthocyanin Reflectance Index | $(1/\rho531) - (1/\rho701)$ | [63] |
| ACI | Anthocyanin Content Index | $(\rho531 - \rho571)/(\rho531 + \rho571)$ | [55] |
| CRI | Carotenoid Reflectance Index | $\rho511/\rho571$ | [64] |
| PR1 | Photochemical Reflectance Index 1 | $\rho529/\rho570$ | [65] |
| RVSI2 | Red-Edge Vegetation Stress Index | $(\rho712 + \rho753)/2 - \rho733$ | [66] |

Subsequently, standard deviation (SD) and coefficient of variation (CV) values were calculated at the plot level, considering all spectral bands and estimated VIs, totaling 606 possible explanatory metrics. SD and CV values were calculated in two ways: (i) considering the entire plot area, including shaded pixels (no mask), and (ii) considering a mask with only illuminated pixels in the plot, excluding shaded ones (mask). A mask of illuminated pixels [67] was created for each plot from NDVI images (NDVI $\leq$ 0.84; red = 674 nm, NIR = 830 nm) using the QGIS Geographic Information System (version 3.40.0) [68] (Figure 2). Pixels with NDVI values below 0.84 were masked to remove non-vegetated or non-foliated areas, as shaded areas are characterized by lower overall reflectance compared to sunlit pixels, particularly in the near-infrared (NIR) domain [69]. These shaded pixels may cause a local reduction in spectroradiometric data, interfering with spectral metrics, especially in densely vegetated environments [70].

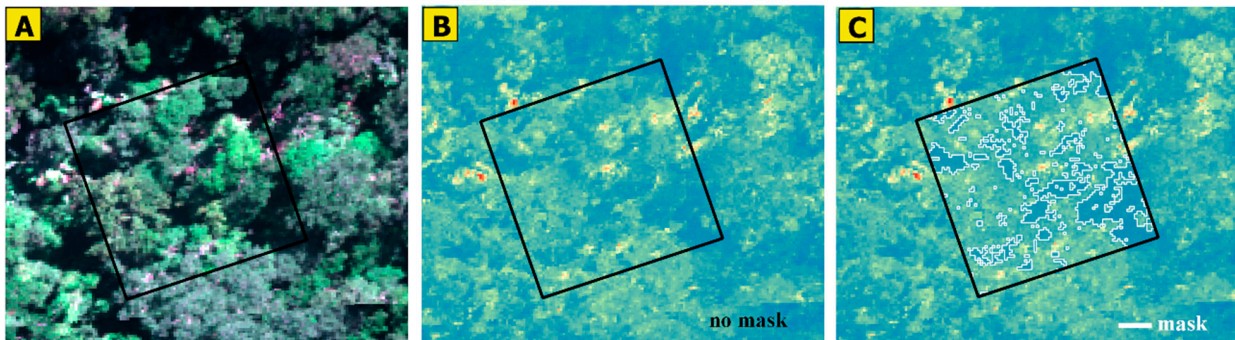

**Figure 2.** Representation of the mask applied to plot 13. (**A**) RGB image collected by the hyperspectral sensor; (**B**) image generated from NDVI without mask; (**C**) image generated from NDVI with the mask applied (NDVI $\leq$ 0.84).

All analyses were performed considering three different spatial resolutions: 1.0, 5.0, and 10.0 m to evaluate the effects of pixel size [8,11,71,72]. The selection of pixel sizes was based on studies emphasizing the importance of adapting spatial resolution to spectral variability and species diversity [8,73–75]. Higher resolutions may increase internal variability,

while lower resolutions may fail to adequately capture diversity. Additionally, the mean canopy diameter (3.8 m) was considered to ensure the spatial resolution was appropriate for the scale of the vegetation.

### 2.4.2. Selection of Hyperspectral Metrics

First, to analyze the relationship between species diversity and richness with the metrics derived from the hyperspectral RPA system (SD and CV) across the electromagnetic spectrum, Pearson correlation coefficients were calculated between species richness at the plot level, H', C, and the reflectance values for each analyzed band. Pearson coefficients were also calculated for vegetation indices (VIs).

Subsequently, the metrics most strongly associated with species richness, H', and C were selected. This selection considered (i) only illuminated pixels (mask) and (ii) shaded pixels (no mask) for each plot. Additionally, spatial resolutions of 1.0, 5.0, and 10.0 m were analyzed. The exhaustive search algorithm from the leaps package [76] in the R programming environment (version 4.4.1) [50] was used for this process.

For each scenario (mask vs. no mask and pixel size), simple linear regression models were then fitted to relate species richness and diversity indices measured in the field with the selected metrics. The model with the lowest Bayesian Information Criterion (BIC) value and the highest coefficient of determination ($R^2$) was selected. The significance of the coefficients in the fitted models was evaluated using the *t*-test at a 5% significance level, and the relationship between richness and diversity parameters with the selected hyperspectral metrics was visualized through scatter plots.

Given the well-established importance of VIs in studies on this topic [12,15,16,26,27,77,78], simple linear regression models were also fitted for the different scenarios to relate richness and diversity parameters to NDVI (SD). The complete process of hyperspectral metric selection is illustrated in the flowchart shown in Figure 3.

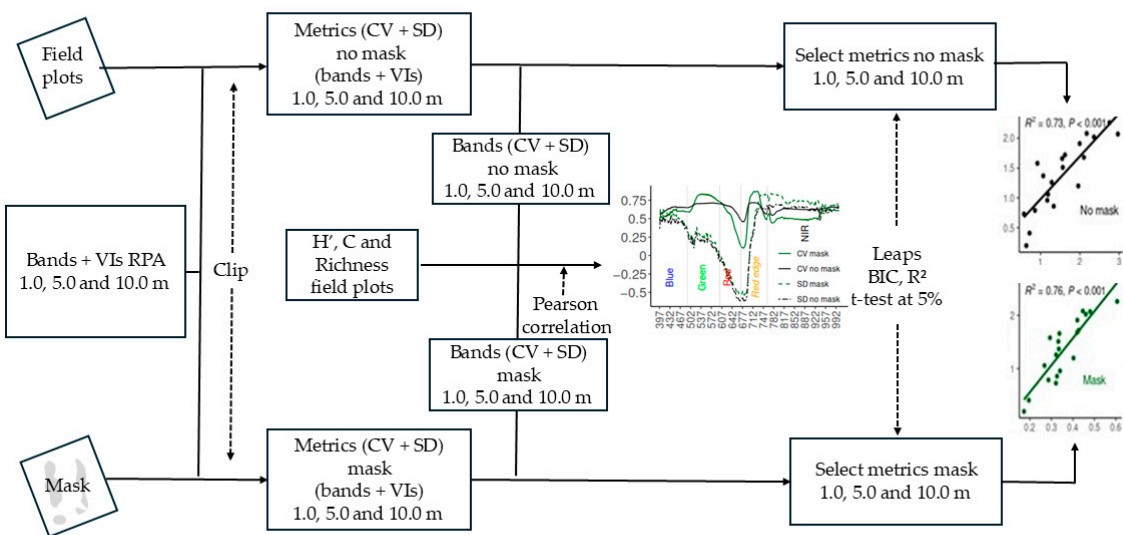

**Figure 3.** Flowchart of the hyperspectral metric selection process (vegetation indices and bands).

### 3. Results

*3.1. Relationship Between Diversity Indices and Spectral Diversity of Bands*

The correlation between the field-measured species richness/diversity and the spectral diversity metrics from the hyperspectral RPA system are shown in Figure 4. Overall, considering the same spatial resolution, we observed similar dynamics for the Shannon (H') and Simpson (C) diversity indices. However, a slightly different relationship was observed for species richness (S) across the spectrum (Figure 4G–I). Reducing the spatial resolution decreased the correlation values and altered their strength across the spectrum.

The negative effect of shaded pixels (black lines "no mask") was evident, often showing lower correlation values compared to illuminated pixels ("mask", green lines).

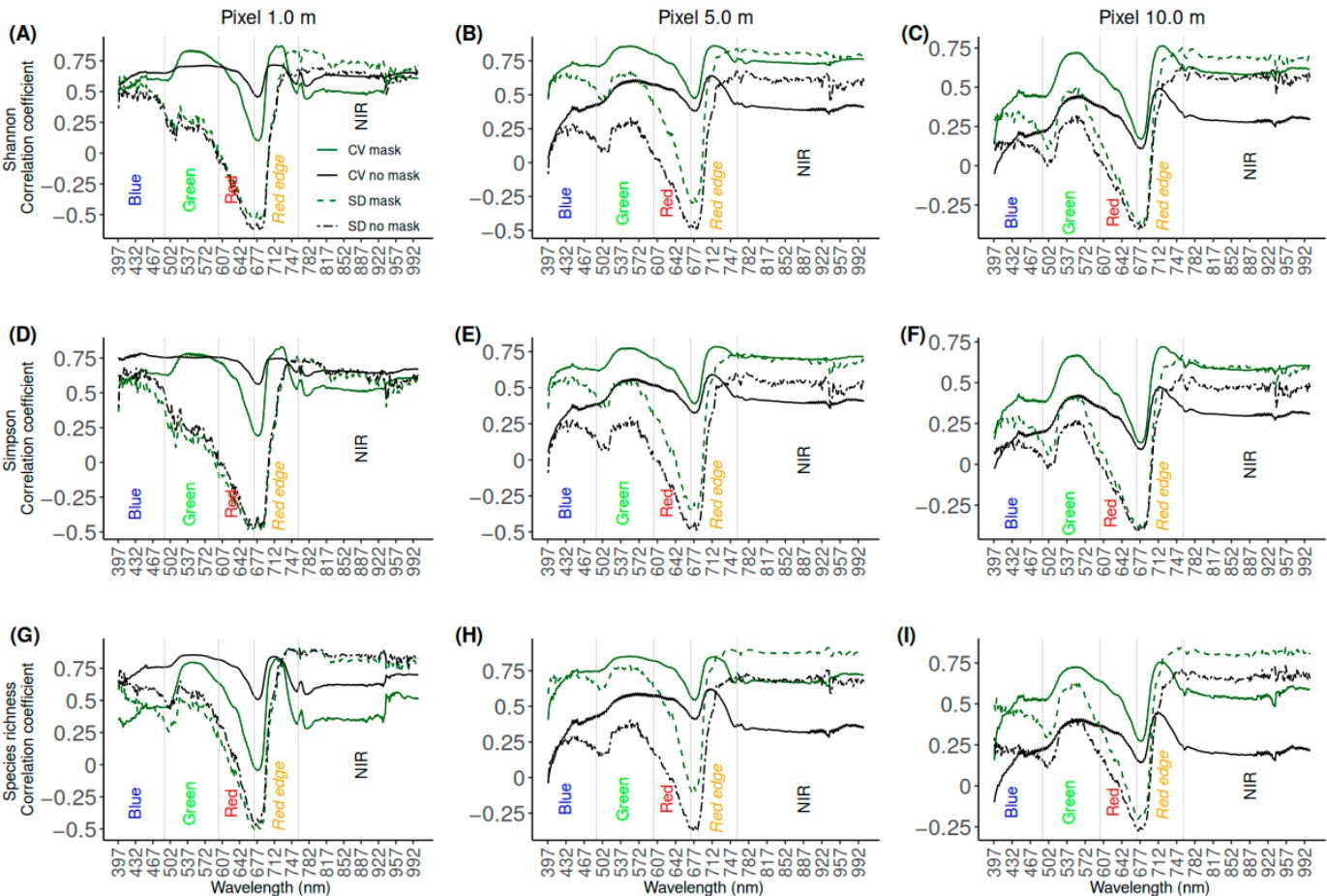

**Figure 4.** Correlation coefficients between field-estimated Shannon (panels (**A**–**C**)), Simpson (panels (**D**–**F**)), and richness (panels (**G**–**I**)) indices and coefficients of variation (CV) and standard deviations (SDs) of reflectance, considering shaded (no mask) and non-shaded (mask) pixels at three different spatial resolutions (1.0, 5.0, and 10.0 m pixels).

Significant changes in correlation were observed throughout the spectrum, regardless of spatial resolution and use of standard deviation (SD) or coefficient of variation (CV). Correlation direction changes were found only for SD values (dashed lines), regardless of resolution and diversity measures, with both direct and inverse correlations. This result differs from the metrics based on CV values (solid lines), which were always positively correlated with the analyzed diversity and richness metrics.

The type of spectral diversity metric also influenced the correlation strength and direction, with the highest correlation values observed for the CV of the Shannon and Simpson indices. The highest correlations ($r$) for species richness were observed for SD values from the spectral bands (Figure 4G–I). The NIR region of the spectrum, considering illuminated pixels, was highly correlated ($r > 0.8$) with species richness, regardless of spatial resolution. The lowest correlations ($r{\sim}0$) for the Shannon and Simpson indices when using 1.0 m pixels (Figure 4A,D) were observed between the green and red bands and in the central part of the red edge. Contrastingly, the highest correlations for these two diversity measures were observed at the end of the red edge, considering CV values, regardless of spatial resolution. The lowest correlations for species richness using 1.0 m pixels were observed in the visible red region and the central part of the red edge (Figure 4G). As

pixel size increased, the lowest correlations were also observed in the visible blue region, regardless of the analyzed diversity and richness measures.

The correlation between the Shannon diversity index and hyperspectral metrics decreased with increasing pixel size, with values ranging from 0.76 to 0.87 for illuminated pixels and 0.63 to 0.85 for shaded pixels (Table 4). Similarly, the correlation was positive for the Simpson index and decreased with increasing pixel size, with values ranging from 0.72 to 0.83 and 0.55 to 0.79 for pixels with and without shadow, respectively (Table 4). Regarding species richness, the correlation with metrics was similar for 1.0 and 5.0 m pixels and decreased at 10.0 m for masked pixels (Table 4). The correlation also reduced with increasing size for shaded pixels.

**Table 4.** Selected hyperspectral metrics to explain field-measured species richness, Shannon and Simpson indices, and correlation coefficient (*r*) values, considering shaded (no mask) and illuminated (mask) pixels at three different spatial resolutions (1.0, 5.0, and 10.0 m pixels).

| Pixel (m) | Mask | Shannon | *r* | Simpson | *r* | Richness | *r* |
|---|---|---|---|---|---|---|---|
| 1 | yes | CV MEAN | 0.87 | CV 726.7 nm | 0.83 | SD 742.3 nm | 0.91 |
| 5 | yes | CV 717.8 nm | 0.86 | CV 717.8 nm | 0.78 | SD 933.4 nm | 0.91 |
| 10 | yes | CV 722.3 nm | 0.76 | CV 722.3 nm | 0.72 | SD 755.6 nm | 0.84 |
| 1 | no | SD C2 | 0.85 | CV 442.3 nm | 0.79 | SD 762.3 nm | 0.90 |
| 5 | no | SD 715.2 nm | 0.69 | SD 757.8 nm | 0.61 | SD 931.1 nm | 0.75 |
| 10 | no | SD 757.8 nm | 0.63 | SD 757.8 nm | 0.55 | SD 955.6 nm | 0.73 |

MEAN = mean ($\rho$691.1 nm to $\rho$740.0 nm); C2 = Chlorophyll Index ($\rho$700 nm/$\rho$751.2 nm); SD = standard deviation; CV = coefficient of variation.

### 3.2. Main Correlated Metrics

In most cases, the individual hyperspectral RPA bands were selected to explain the H', C, and S values (Table 4 and Figure 5). Additionally, the Chlorophyll Index (C2) and mean reflectance values ranging from 690.0 to 740.0 nm (red edge) were selected for the Shannon index when using 1.0 m pixels (Table 4 and Figure 5A,B).

The hyperspectral metrics selected for the Shannon index were CVMEAN and SD C2, both in the red edge for illuminated and shaded 1.0 m pixels. The metrics with the highest correlation values were CV 726.7 (red edge) and CV 442.3 (blue), with r = 0.83 for masked pixels and r = 0.79 for unmasked pixels. The hyperspectral metrics selected for species richness were SD 742.3 nm (red edge) and SD 933.4 (NIR) for illuminated 1.0 and 5.0 m pixels, respectively, with a strong correlation (0.91). The selected metric for shaded pixels was SD 762.3 nm (red edge), with a value of 0.90.

All selected metrics were significant at a 5% significance level (Figure 5) and showed correlations that could be classified as moderate (*r* ~ 0.55) to strong (*r* > 0.91) (Table 4). The highest correlations were observed for species richness, followed by H' and C. Most selected bands were in the red-edge region of the electromagnetic spectrum (~670 to 760 nm), confirming the importance of this range in spectral diversity studies. Pixel size increase reduced the correlation, even for metrics extracted from illuminated (mask) and shaded (no mask) pixels.

The negative effect of shaded pixels (no mask) on the extraction of dispersion metrics was evident, as shown in Figure 4. The *r* value was consistently lower compared to illuminated pixels (mask) across all analyzed spatial resolutions. Regarding the illuminated pixels, the selection of spectral dispersion measures is based on the CV values for H' and C, regardless of the pixel size. However, spectral metrics derived from SD were selected for species richness values, regardless of spatial resolution or shadow effects on the plot pixels. The highest correlations were observed for the richness metric (*r* > 0.73). When considering metrics extracted from shaded pixels, SD-based dispersion measures were also selected, except for C at the 1.0 m spatial resolution (Table 4 and Figure 5H).

Scatterplots showing the relationship between diversity/richness values and the selected hyperspectral RPA metrics from Table 4 are shown in Figure 5. Overall, reducing

the spatial resolution increased data dispersion, as observed for 5.0 and 10.0 m resolutions. The same pattern was observed when considering the effects of shaded pixels (black dots and lines). The relationships between diversity indices and the presented metrics were significant ($p < 0.05$) by the *t*-test.

The $R^2$ values were higher for H' at 1.0 m pixels, with 0.76 and 0.73 for masked and unmasked pixels, respectively, and decreased as pixel size increased (first row of Figure 5). The same was found for the Simpson index (Figure 5), with the best values also obtained when using 1.0 m pixels, with 0.69 for illuminated pixels and 0.62 for shaded pixels. The highest $R^2$ values were found for richness across all pixel sizes, with or without shadow effects, compared to diversity indices, reaching 0.83 for illuminated 1.0 and 5.0 m pixels (Figure 5).

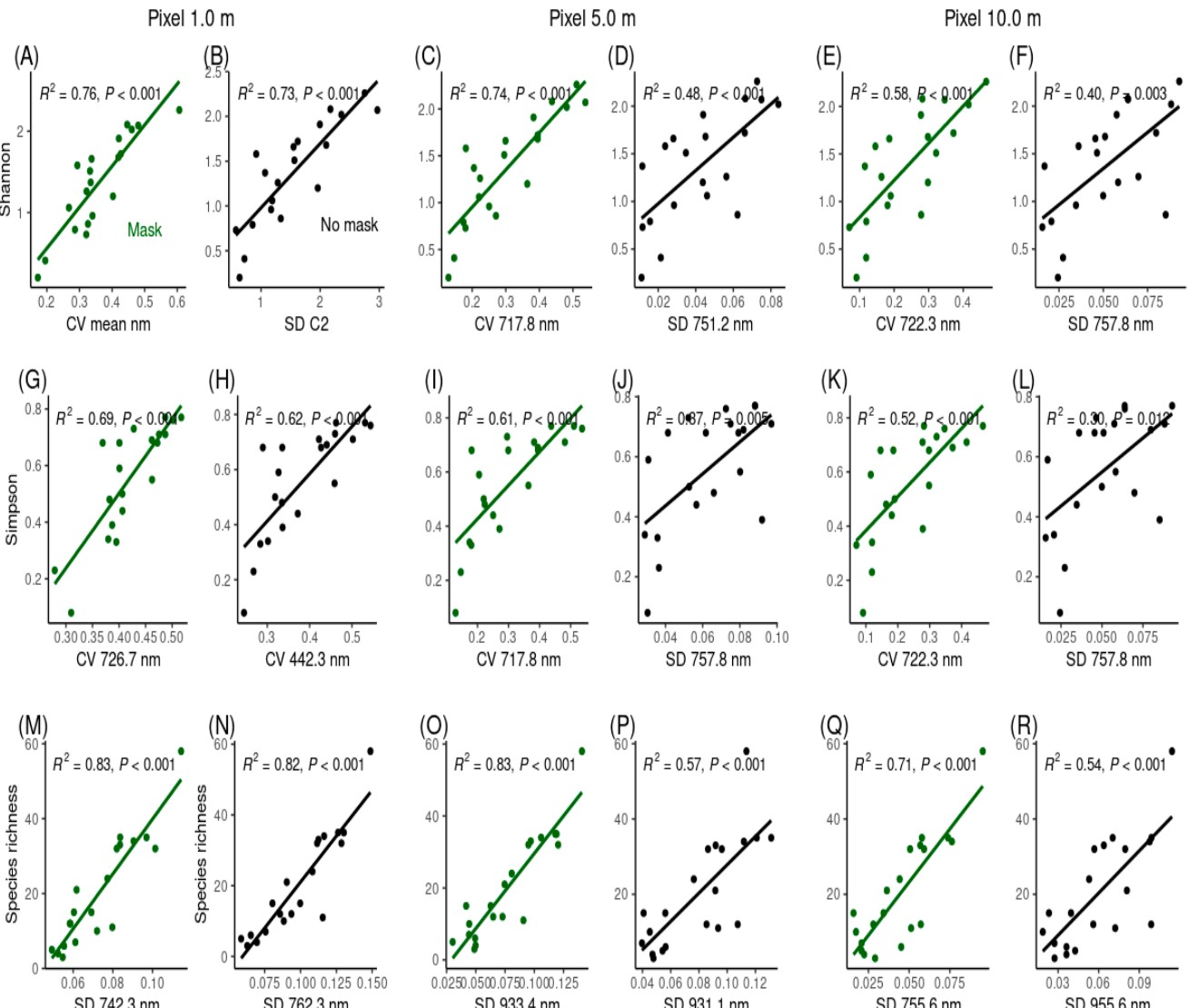

**Figure 5.** Relationship between field-measured Shannon (**A–F**), Simpson (**G–L**), and species richness (**M–R**) and the selected hyperspectral metrics, considering illuminated and shaded pixels in the plot at three different spatial resolutions (1.0, 5.0, and 10.0 m pixels).

### 3.3. Relationship Between Diversity Indices and Vegetation Indices

Although the vegetation indices (VIs) were selected only to explain the Shannon values for 1.0 m pixels (Figure 5A,B), they showed a strong correlation with H', C, and S values

(Tables 5 and 6). Moderate negative correlations ($r < -0.49$) were found for all diversity values analyzed.

Similar to individual spectral bands (Table 4 and Figure 5), the correlation decreased as pixel size increased. The same shadow effect was observed, with lower correlation values. Overall, the CV values of VIs performed better for illuminated pixels (Table 5), differing from extractions based on both illuminated and shaded pixels (Table 6), where VIs extracted based on SD values were more strongly correlated. The VI most correlated with taxonomic diversity varied depending on pixel size, with an emphasis on CV values of MEDI, RVSI, and PRI for illuminated pixels (Table 5). SD values extracted from C2, PSSR, PRI, and SR stand out for shaded pixels (Table 6).

NDVI dispersion measures did not show the highest correlations and were not selected (Table 4 and Figure 5) to explain the investigated diversity values. Furthermore, as observed for other VIs, NDVI SD showed negative correlations with diversity measures (H', C, and S) (Figure 6).

**Table 5.** Highest positive and negative correlation coefficients (*r*) between vegetation indices and dispersion metrics (standard deviation—SD or coefficient of variation—CV) extracted from the illuminated pixels of a hyperspectral image.

| Pixel (m) | Shannon | *r* | Simpson | *r* | Richness | *r* |
|---|---|---|---|---|---|---|
| 1 | CV MEAN | 0.87 | CV MEAN | 0.79 | CV MEAN | 0.90 |
| | CV MEDI | 0.86 | CV RVSI2 | 0.79 | CV MEDI | 0.90 |
| | CV RVSI2 | 0.83 | CV MEDI | 0.77 | SD PRI | 0.88 |
| | SD mARI | 0.78 | SD C2 | 0.73 | SD PRI2 | 0.88 |
| | SDCRI | 0.77 | SD PSSR2 | 0.65 | SD mARI | 0.87 |
| | CV CI.green | −0.86 | CV CI.green | −0.79 | CV CI.green | −0.87 |
| | CV PRI | −0.78 | CV CI.rededge | −0.70 | CV PRI | −0.83 |
| | CV PRI2 | −0.78 | CV PRI | −0.67 | CV PRI2 | −0.83 |
| | CV CI.rededge | −0.73 | CV PRI2 | −0.67 | SD NDVI | −0.70 |
| | SD NDVI2 | −0.63 | CV ACI | −0.53 | SD NDVI2 | −0.70 |
| 5 | CV MEAN | 0.78 | CV RVSI2 | 0.71 | CV MEDI | 0.82 |
| | CV MEDI | 0.78 | CV MEAN | 0.70 | SD PRI | 0.81 |
| | CV RVSI2 | 0.77 | CV MEDI | 0.70 | SD PRI2 | 0.81 |
| | SD RVSI2 | 0.76 | CV CARI | 0.70 | CV MEAN | 0.81 |
| | CV CARI | 0.76 | SD RVSI2 | 0.67 | SD RVSI2 | 0.80 |
| | CV CI.green | −0.77 | CV CI.green | −0.70 | CV CI.green | −0.79 |
| | CV CI.rededge | −0.71 | CV CI.rededge | −0.67 | CV PRI | −0.79 |
| | CV PRI | −0.68 | CV ACI | −0.60 | CV PRI2 | −0.79 |
| | CV PRI2 | −0.68 | CV PRI | −0.55 | CV CI.rededge | −0.68 |
| | CV ACI | −0.60 | CV PRI2 | −0.55 | SD NDVI2 | −0.61 |
| 10 | SD RVSI2 | 0.64 | CV CARI | 0.58 | SD ACI | 0.74 |
| | CV CARI | 0.63 | SD RVSI2 | 0.55 | SD PRI | 0.74 |
| | SD PRI | 0.63 | CV MEDI | 0.55 | SD PRI2 | 0.74 |
| | SD PRI2 | 0.63 | CV RVSI2 | 0.55 | SD RVSI2 | 0.73 |
| | SD ACI | 0.62 | CV MEAN | 0.54 | SD CI.rededge | 0.72 |
| | CV PRI | −0.63 | CV CI.green | −0.54 | CV PRI | −0.69 |
| | CV PRI2 | −0.63 | CV PRI | −0.52 | CV PRI2 | −0.69 |
| | CV CI.green | −0.57 | CV PRI2 | −0.52 | CV CI.green | −0.54 |
| | CV CI.rededge | −0.50 | CV CI.rededge | −0.48 | CV NDVI | −0.50 |
| | CV SIPI | −0.49 | CV SIPI | −0.47 | CV NDVI2 | −0.50 |

**Table 6.** Highest positive and negative correlation coefficients (r) between vegetation indices and dispersion metrics (standard deviation—SD or coefficient of variation—CV) extracted from all pixels of a hyperspectral image.

| Pixel (m) | Shannon | r | Simpson | r | Richness | r |
|---|---|---|---|---|---|---|
| 1 | SD C2 | 0.85 | SD C2 | 0.78 | SD PRI | 0.83 |
| | SD PSSR2 | 0.83 | SD PSSR2 | 0.75 | SD PRI2 | 0.83 |
| | SD PRI | 0.82 | SD PRI | 0.73 | SD C2 | 0.81 |
| | SD PRI2 | 0.82 | SD PRI2 | 0.73 | SD CRI | 0.81 |
| | SD SR | 0.80 | SD PSSR1 | 0.73 | SD PSSR2 | 0.79 |
| | CV PRI | −0.70 | CV PRI | −0.61 | SD NDVI | −0.69 |
| | CV PRI2 | −0.70 | CV PRI2 | −0.61 | SD NDVI2 | −0.69 |
| | SD NDVI2 | −0.62 | CV CI.green | −0.58 | SD PSI | −0.68 |
| | SD NDVI | −0.62 | SD ACI2 | −0.52 | CV NDVI | −0.68 |
| | SD PSI | −0.61 | SD NDVI2 | −0.52 | CV NDVI2 | −0.68 |
| 5 | SD C2 | 0.66 | SD C2 | 0.58 | SD ACI | 0.69 |
| | SD PSSR2 | 0.65 | SD PSSR2 | 0.57 | SD RVSI2 | 0.68 |
| | SD PRI | 0.65 | SD RVSI2 | 0.56 | SD CI.rededge | 0.68 |
| | SD PRI2 | 0.65 | SD CI.rededge | 0.56 | SD PRI | 0.67 |
| | SD CRI | 0.65 | SD PRI | 0.55 | SD PRI2 | 0.67 |
| | CV PRI | −0.59 | CV CI.green | −0.51 | CV PRI | −0.59 |
| | CV PRI2 | −0.59 | CV PRI | −0.49 | CV PRI2 | −0.59 |
| | CV CI.green | −0.56 | CV PRI2 | −0.49 | CV NDVI2 | −0.57 |
| | CV CI.rededge | −0.50 | CV CI.rededge | −0.47 | SD PSI | −0.57 |
| | SD PSI | −0.49 | SD SIPI | −0.41 | CV NDVI | −0.57 |
| 10 | SD ACI | 0.61 | SD RVSI2 | 0.52 | SD ACI | 0.71 |
| | SD PRI | 0.60 | SD CI.rededge | 0.50 | SD PRI | 0.69 |
| | SD PRI2 | 0.60 | SD ACI | 0.50 | SD PRI2 | 0.69 |
| | SD CI.rededge | 0.59 | SD PRI | 0.49 | SD CI.rededge | 0.67 |
| | SD RVSI2 | 0.59 | SD PRI2 | 0.49 | SD RVSI2 | 0.65 |
| | CV PRI | −0.60 | CV PRI | −0.49 | CV PRI | −0.63 |
| | CV PRI2 | −0.60 | CV PRI2 | −0.49 | CV PRI2 | −0.63 |
| | SD R1 | −0.43 | SD R1 | −0.37 | SD NDVI | −0.53 |
| | SD PSI | −0.43 | SD PSI | −0.36 | SD R1 | −0.53 |
| | SD NDVI2 | −0.42 | CV SIPI | −0.34 | SD NDVI2 | −0.53 |

The correlation was consistently negative for NDVI (Figure 6), ranging from moderate to weak ($R^2 < 0.46$) across all analyzed spatial resolutions, even when considering SD values of illuminated (mask, green color) and shaded pixels (no mask, black color). Additionally, a reduction in correlation was observed with increasing pixel size. Shaded pixels did not influence the correlation at the highest spatial resolution (1.0 m) (Figure 6A,D,G) or the data dispersion. Furthermore, all correlations were significant by the t-test ($p < 0.05$), and the best result was observed for the correlation between NDVI and species richness (Figure 6G–I).

For the other spatial resolutions (5.0 and 10.0 m), a reduction in correlation increased data dispersion, and the shadowing effects on pixels were observed. Considering the Shannon index and species richness at the 5.0 m resolution (Figure 6B,H), despite the weak correlation, the correlations were significant by the t-test ($p < 0.05$), unlike the Simpson index ($p > 0.05$—Figure 6E). At the 10.0 m resolution, most correlations were not significant ($p > 0.05$) (Figure 6C,F).

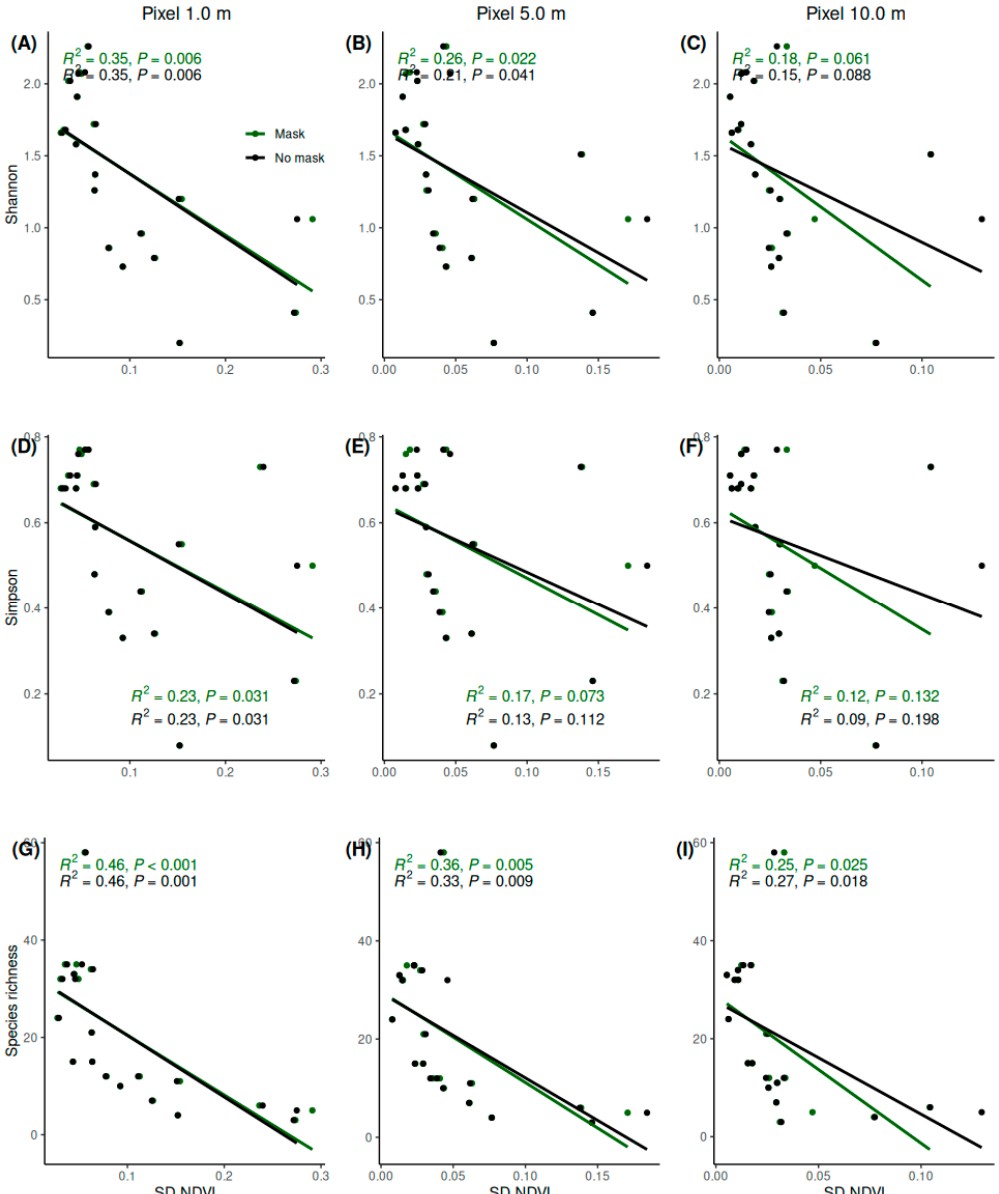

**Figure 6.** Relationship between field-estimated Shannon (**A–C**), Simpson (**D–F**), and species richness (**G–I**) and NDVI SD values, considering shaded and illuminated pixels of the plot at three different spatial resolutions (1.0, 5.0, and 10.0 m pixels), with Shannon and Simpson indices calculated based on coverage.

## 4. Discussion

### 4.1. Relationship Between Diversity Indices and Spectral Diversity

The Spectral Variation Hypothesis (SVH) was confirmed for the Atlantic Forest biome, but it can be influenced by several factors, such as the selected metrics, which directly depend on the sensor used, pixel size, and shadow effects. The SVH is based on the high diversity of this biome, which results in significant variations in spectral reflectance from different vegetation types and environmental conditions. Spectral variations captured by remote sensing technologies are important tools for identifying and monitoring the structural complexity and biological diversity of the Atlantic Forest, reinforcing the usefulness of spectral mapping for studies on conservation and sustainable management in one of the world's richest and most threatened ecosystems [32].

The coefficient of variation (CV) was the dispersion measure most strongly associated with taxonomic diversity in the red-edge region. The red edge is highly sensitive to

changes in species composition and leaf properties, such as chlorophyll content [79], and represents a sharp transition in vegetation reflectance, making CV a better measure of relative variation in this region. Contrastingly, the standard deviation (SD), which measures the absolute dispersion of data around the mean, was the strongest measure correlated with diversity in the NIR. In this region, vegetation reflectance is generally high and more uniform, influenced by the internal leaf structure that varies among species and health conditions [80], and the mean reflectance tends to be more consistent, reducing the need for normalization provided by the CV. In this context, SD captures the absolute variations in reflectance that are indicative of the structural and compositional diversity of vegetation. This is likely why SD showed a stronger correlation with species diversity in the NIR in the present study, as it reflects the structural variations in plants, which are significant in this spectral region.

Most studies relating spectral diversity to taxonomic diversity using hyperspectral sensors also found the highest correlation values for the red-edge and NIR regions [81–83]. The best spectral range for metric selection varies among studies, depending on habitat heterogeneity and sensors used. Reference [83] mapped mangrove forests in China using multispectral and hyperspectral imagery (WorldView-2, Sentinel-2, and Zhuhai-1) and found that spectral characteristics of the red edge and NIR were the most informative for diversity mapping ($R^2$ = 0.2 to 0.42), particularly shortwave infrared, which was also valuable for beta diversity (β) mapping.

Similarly, [82] found the highest correlation (pseudo-$R^2$ = 84.9%) of species richness and Shannon diversity (H') with the SD of reflectance in the NIR region in a tropical forest in West Africa; the study also selected metrics with high correlation values ($R^2$ = 0.83). These results can be explained by the high reflectance in longer wavelengths of the NIR (700–1100 nm) region, and the influence of leaf structural characteristics (e.g., intercellular spaces and cell wall thickness) and canopy architecture (e.g., branching structure and leaf angle distribution) on light scattering [80].

A higher correlation of Shannon (H'), Simpson (C), and species richness (S) indices with longer wavelengths in the NIR was also observed by [81]. They used two imaging spectrometers mounted on a fixed-wing aircraft to acquire hyperspectral data in temperate and boreal forests in Canada. Leaf pigments exhibit strong absorption characteristics in the 700–1100 nm range [80], and pigment content differs between deciduous broadleaf and coniferous species. Thus, there is a higher correlation of species diversity with spectral diversity in wavelengths within the 700–1100 nm range.

Species richness, Shannon, and Simpson indices are widely used as diversity indices in the remote sensing literature [12,15,16,84], but their correlations with band dispersion variables and spectral indices vary depending on the study. In the present research, within the same spatial resolution and spectral dispersion metric (CV or SD), the analyzed diversity measures showed varied dynamics for Shannon, Simpson, and richness indices across the considered electromagnetic spectrum range; however, overall, species richness was more strongly correlated with spectral variability than H' and C diversity.

In wetlands in China, [15] used high-resolution multispectral imagery to correlate commonly used species diversity indices, such as species richness, Shannon, and Simpson, calculated from field data. The results showed that all indices had the highest predictive capacity for species richness, followed by Shannon, and the lowest for Simpson, with $R^2$ values of 0.45, 0.37, and 0.32, respectively, when correlated with NDVI. Despite differences in study locations, vegetation types, sensors, and related metrics, the $R^2$ values, although lower, followed a similar trend to those found in the present study, with higher values for species richness, Shannon, and Simpson indices. In another study, using coverage instead of density to calculate diversity indices, but in the same location and with the same sensor, [16] found that the ability to predict taxonomic diversity from NDVI was the same for species richness, Shannon, and Simpson indices, with $R^2$ = 0.45.

Species richness is the most widely used measure of biodiversity [85] and has been related to spectral diversity metrics in most SVH studies [11,12,15,16,22,26]. However, [82]

argued that the Shannon index is more valuable from an ecological perspective because it better reflects the structural variability of a landscape, as it captures differences in the composition and dominance structure of the plant community. According to these authors, the Shannon index is calculated from species richness and density/abundance measures, making it more useful for relating spectral signals to local ecological processes. Reference [22] reported that H' and C indices, compared to species richness, better represent vegetation structure, which is a subset of habitat heterogeneity and, therefore, should better reflect spectral variability. However, this was not influential in the present study, as better correlations were obtained between spectral diversity and richness ($R^2$ = 0.83), and significant values were also found for the Shannon and Simpson indices ($R^2$ = 0.76 and 0.69, respectively).

The H' index considers both species richness and evenness, making it more sensitive to changes in species composition [4]. Spectral diversity can capture subtle variations among different species, reflecting richness and evenness better than the C index, which is more influenced by the dominance of a few species. Spectral diversity can detect the presence of rare species and functional variations that are important for richness and the H' index. Contrastingly, the C index is less sensitive to rare species and more focused on dominance [4], which may explain the lower correlation found between Simpson index values and spectral diversity in the present research. Furthermore, the Simpson diversity index is less sensitive to species richness than Shannon index, as the link between spectral variation and species diversity is partly based on interspecies variation in spectrum and leaf structure, which may reduce the performance of the Simpson diversity index in terms of variation in spectral images [23].

Similar results regarding the correlation between spectral and taxonomic diversity were found in other works involving hyperspectral data. Reference [86] applied the "spectronomic" method combined with airborne hyperspectral data (PHI-3 sensor with a spatial resolution of 1 m) and LiDAR (>4 points m$^{-2}$) to identify interspecies variations in biochemical and structural properties of trees and then estimate tree species diversity in China. Species richness and the Shannon–Wiener diversity index calculated from clustering results correlated well with field reference data ($R^2$ = 0.83, RMSE = 0.25). Reference [87] used AVIRIS data to characterize species diversity and forest health in India and concluded that these data could be used for species delineation and community-level diversity mapping. However, the accuracy achieved in species classification is moderate (60%) in the studied site due to the predominance and mixture of species in the area. According to the authors, AVIRIS-NG data with higher spectral and spatial resolutions provided a unique opportunity to perform biochemical-level spectroscopy of the forest landscape, which was not possible with the available multispectral observations.

Studies have shown that the relationship between diversity indices and spectral band diversity is influenced by various factors, such as spatial resolution effects, effects stemming from the size of field plots, zoning or selection of investigation areas, spectral resolution, and the timing of the investigation, as well as the location and extent of the reference region [9,88]. Furthermore, habitats that may appear spectrally very similar due to the presence of dominant canopy species can still differ widely in species count due to species present beneath the canopy [8], which can influence correlation results obtained by different indices.

Regarding the main metrics selected, a significant portion of the most highly correlated indices and bands in the present study were in the red edge (670 to 760 nm) of the electromagnetic spectrum, where there is an abrupt transition from low reflectance (in the red range) to high reflectance (in the near-infrared range) in plant leaves. The red-edge region is extremely sensitive to changes in vegetation composition and structure. Variations in reflectance in this spectral region allow for the discrimination of different plant species with greater precision than other bands [79]. In addition to plant species discrimination, the red edge allows for the identification of their ecosystem functions due to its unique spectral

properties that reflect differences in the chemical composition and physical structure of plants [89].

The red-edge and NIR bands, along with associated indices, have been successfully used to predict not only species diversity but also leaf area indices, vegetation structure, and tree species distribution [90].

These bands are linked to the biochemical concentrations and spectral characteristics of foliar chemistry, providing information on water content, chlorophyll pigments, carotenoids, nitrogen, and other components [91]. Spectral variations, induced by biochemical and biophysical parameters of the canopy, are generally associated with leaves and are known as functional traits due to their specific roles within the plant [92]. Photosynthetic pigments (chlorophylls, carotenoids, anthocyanins, etc.) and nitrogen contribute to plant growth, while cellulose and lignin are responsible for leaf structure.

Studies indicate that functional traits in tropical forests are taxonomically organized [93] or directly related to species composition [94], suggesting that their monitoring may provide valuable information on functional diversity [95]. Additionally, the chemical attributes of leaves are associated with functional diversity and can be used to differentiate species [92].

Specifically, the bands 717 nm, 722 nm, 715 nm, and 742 nm, within the red-edge range, exhibit sharp transitions from absorption to reflectance in leaves, allowing precise species discrimination and a robust correlation with taxonomic diversity [32,96–98]. The red-edge and green spectral ranges are particularly sensitive to photosynthetic pigment content, especially in dense canopies, reflecting a connection between taxonomic information, photosynthetic activity, and spectral data [99]. In contrast, blue and red bands, which correspond to chlorophyll absorption domains, have shown a negative contribution to the relationship between taxonomic diversity and spectral variance, as they tend to saturate in dense, highly photosynthetic canopies [69].

The NIR region (740–1100 nm) demonstrated a strong correlation with species diversity due to its ability to capture variations in the optical properties of leaves, which depend on internal cell structure, and in canopy architecture, both essential for species differentiation [100]. The bands 757 nm, 726 nm, 755 nm, 762 nm, 931 nm, 933 nm, and 955 nm, within the NIR and SWIR ranges, are associated with leaf reflectance and vegetation structure. These bands are influenced by leaf cellular structure and canopy density, which vary among species and indicate structural and functional diversity [32,96–98].

### 4.2. Effects of Pixel Size on the Relationship Between Spectral Diversity and Taxonomic Diversity

Overall, the reduction in spatial resolution (pixels of 5.0 and 10.0 m) resulted in a significant decrease in correlations across the spectrum, with a more pronounced impact in the red-edge and NIR regions, which are particularly sensitive to the structural and biochemical properties of vegetation [8,32,96–98]. This reduction was especially evident in the coefficient of variation (CV) and standard deviation (SD) values estimated without the application of illuminated pixel masks. At coarser resolutions (5.0 m and 10.0 m), a notable decrease in correlation values was observed across all spectral regions. This effect was most pronounced in the red-edge region, where the sharp transition from low to high reflectance characteristic of vegetation is smoothed by pixel aggregation, reducing the ability to detect fine-scale spectral variations associated with species diversity. This "smoothing effect," induced by larger pixel sizes, dilutes spectral details and diminishes the ability to capture ecological patterns at fine scales, as described by [8].

Studies have shown that pixel size can be a crucial factor in spectral variability and taxonomic diversity studies, as it influences the ability to detect and differentiate species and their spectral characteristics [8]. The need for a particular spatial resolution varies according to the vegetation type. In secondary tropical forests, like those in the present study, smaller pixels are needed to differentiate individual tree canopies, while larger pixels may suffice in savannas or agricultural areas [101].

The effects of spatial and spectral resolution were tested by [11] using Aster, Landsat ETM+, and Quickbird images to estimate species diversity in Italy. They reported that

spectral variability depends on the scene and sensor, but low-spatial-resolution images showed very low power in capturing spectral variability due to mixed-pixel problems, which are less sensitive to spatial complexity. Smaller pixels capture more details of spatial heterogeneity within an ecosystem, allowing for the detection of fine-scale vegetation variations, which is essential for identifying individual species or small species groups [102].

Spatial resolution influences the accuracy of calculated diversity indices. Smaller pixels allow for a more precise assessment of species composition and structural diversity [103]. According to [8], the lower the image resolution, the lower the overall spectral variation across pixels in an area, as data aggregation reduces extreme values (smoothing effect). Thus, it can be assumed that any spectral variation metric will decrease with increasing pixel size. High-spatial-resolution data are more likely to differentiate spectral responses of individual species and capture land cover types or individual objects with extreme spectral dynamics.

Our study highlighted that reducing spatial resolution negatively impacted the correlation between spectral diversity and taxonomic diversity, with $R^2$ values decreasing as pixel size increased from 1.0 m to 10.0 m. For illuminated pixels, the highest $R^2$ value obtained was 0.83 for species richness at 1.0 m, while lower values were recorded for coarser resolutions (5.0 m and 10.0 m). These results are consistent with those reported by [11], who found *r* values of 0.69 for QuickBird, decreasing to 0.43 and 0.67 for Aster and Landsat, respectively, demonstrating that higher-resolution sensors better capture spectral variability associated with species richness.

Limitations in capturing spectral variability were also observed for the 10 m resolution of Sentinel-2 and 30 m of DESIS, with $R^2$ values around 0.89 in localized simulations [13]. The study concluded that spatial resolution is a significant limitation. These $R^2$ values indicate that even moderate resolutions can constrain accuracy in areas with high spectral heterogeneity, such as secondary tropical forests. However, the stronger correlations observed in our study at 1.0 m highlight the importance of fine spatial resolutions for capturing small-scale variations in complex ecosystems.

Resolutions ranging from 2.4 m (QuickBird) to 30 m (Landsat ETM+ and ASTER) were also investigated, achieving $R^2$ values of up to 0.87 for mean NDVI correlated with species richness across altitudinal gradients [26]. These findings support the notion that finer spatial resolution is crucial for detecting differences in species composition, particularly in areas with dense vegetation and complex structural layers.

However, [104], who used Sentinel-2 (10–20 m) and Landsat-8 (30 m) in alpine forests, reported $R^2$ values of up to 0.70 for NDVI, highlighting that the suitability of spatial resolution also depends on the habitat type and spectral metric used. This contrast suggests that less dense habitats, such as savannas or agricultural areas, may not require resolutions as fine as those needed for tropical forests.

The findings of [17] further support this perspective by showing that vegetation cover explained between 53% and 84% of the variance in models based on Sentinel-2 (20 m), indicating that lower resolutions may be more effective in homogeneous areas. However, the secondary tropical forests analyzed in our study, which exhibit high structural and compositional diversity, require finer resolutions to capture spectral variations related to species diversity.

Finally, the impact of pixel size on $R^2$ values was also influenced by shadow effects. Shadowed data consistently reduced correlations across all analyzed pixel sizes, with the greatest impact observed at finer resolutions (1.0 m). This effect is consistent with [105], who found reduced correlations when mixed pixels were included in semi-natural grasslands. Our use of illuminated pixel masks helped mitigate this issue, increasing $R^2$ values across all resolutions, which is essential in dense forests where shadow interference is unavoidable.

### 4.3. Effects of Shadow on the Relationship Between Spectral Diversity and Taxonomic Diversity

Shadows negatively impacted correlation coefficients across all spectral regions, with the greatest effect observed in the visible range (blue, green, and red), where reduced

reflectance masks spectral signals associated with species differentiation. In the *red-edge* region (670–760 nm), shadow effects also significantly reduced correlation, whereas sunlit pixels maintained high coefficients, emphasizing the critical role of this region in species discrimination and successional stage identification. In the NIR range (740–1100 nm), correlation coefficients were the highest, capturing structural information related to leaf arrangement and canopy density. Although shadows diminished their effectiveness, the impact was less pronounced compared to other spectral regions.

The studied Atlantic Forest region is characterized by different canopy layers, with a dominant canopy of tall trees and an understory of suppressed or co-dominant trees. Even when pixels of understory trees are sampled by hyperspectral imagery, dominant trees cast shadows on portions of the understory, contributing to pixel overlap (Figure 2). Shadowing affected the strength and direction of the correlation. Shadows cast by tall trees can obscure the underlying vegetation, affecting species detection and diversity estimation [106].

Shaded regions are readily identifiable in high-resolution imagery and significantly affect analytical techniques by altering surface appearance, often leading to the loss of information in covered areas [107]. Shadow is an important factor in studies relating spectral diversity to species diversity because it can introduce variability and noise to spectral data, affecting the accuracy of the analysis [105], as it reduces reflectance in various regions of the spectrum, especially in the visible and NIR bands. This can mask species-specific spectral signatures, challenging identification and classification [108]. Additionally, it can introduce spurious variability into the spectral data, unrelated to species characteristics, due to lighting conditions and acquisition geometry [109].

Sunlit canopy pixels avoid the influence of non-photosynthetic elements of the canopy (e.g., branches and twigs) on spectral diversity quantification. Non-photosynthetic vegetation causes variations in spectral amplitude, i.e., differences in brightness that can increase spectral variability even when spectral shapes are the same [32]. Additionally, pixels affected by shadows may have their spectral diversity underestimated, as shaded areas can appear homogeneous or similar to each other, regardless of the actual differences in species diversity present [98].

In our study, although the Shannon and Simpson indices showed significant correlations with spectral diversity, the correlation values for species richness were slightly higher, particularly in sunlit pixels (mask). However, it was observed that the correlation between species richness and spectral diversity was more affected by shadow effects compared to the diversity indices. This is likely because the indices respond differently to the presence and distribution of species within an area. Species richness simply counts the number of species present without considering their abundance or cover [22,23,84]. Shadows can obscure smaller or less common species, reducing the observed species count and directly lowering richness measurements. When shadows cover parts of the vegetation, they can completely mask certain species, leading to undercounting, especially in highly diverse or structurally complex ecosystems like forests.

On the other hand, both the Shannon and Simpson indices account not only for species presence but also for their abundance or cover [3]. The Shannon index is influenced by species richness and evenness, while the Simpson index gives greater weight to dominant species [22,23,84]. Because these indices incorporate abundance or cover, they are less affected by the potential underrepresentation of less visible species caused by shadows. Shaded areas may not significantly alter these indices if the dominant species remain visible and proportionately accounted for.

### 4.4. Correlation Between Taxonomic Diversity and NDVI

Most studies involving the SVH have used NDVI as a spectral diversity metric to test its correlation with taxonomic diversity [12,15,16,26,27,85,105,110,111]. However, these studies were conducted with multispectral sensors, which have few bands.

NDVI is one of the most widely used remote sensing vegetation indices to quantify ecosystem biomass [15,84]. It is related to energy exchange in an ecosystem and primary

productivity [112], and this relationship has been considered a valid indicator of regional variation in species diversity [104]. Some studies have reported that NDVI variability is related to species diversity [112–114].

Hundreds of bands and a large number of VIs were estimated and tested in the present study, as it was conducted with a hyperspectral sensor. NDVI was not selected as the main spectral diversity metric, possibly because the high spectral resolution of our study allowed for the selection of bands highly sensitive to vegetation properties, such as plant starch, leaf water content, canopy density, and structural carbohydrates [102,115]. Nonetheless, as the most widely studied VIs, the correlation values of NDVI dispersion measures were calculated for comparison with other studies. Unlike other studies that indicate a positive correlation between NDVI variation and species richness and diversity [26,85,110], the correlation of NDVI SD with Shannon, Simpson, and species richness indices was negative or null, with higher values when using 1.0 m pixels, whether illuminated or shaded, but still considered low.

The negative correlation of NDVI SD with species diversity and richness indices can be explained by various ecological and methodological factors, including environmental heterogeneity. Regions with high species diversity often exhibit greater environmental heterogeneity, resulting in a wider range of NDVI values. However, when the vegetation is uniformly distributed in terms of cover and vigor, the NDVI SD may be low even with high diversity. Additionally, in tropical forest ecosystems, which have high species diversity, such as the Atlantic Forest, the vegetation cover can be dense and uniform, leading to consistent NDVI values and, therefore, a lower SD [79], even with high diversity and richness.

A negative correlation between species richness and NDVI SD was found by [114] using Landsat satellite images in Florida, USA, which can be explained by the dominance of late-successional species, reducing NDVI variation. In pastures in the Czech Republic, [28] used a multispectral sensor on an RPA and found that taxonomic diversity was also negatively related to spectral diversity, which was unexpected. This was probably due to the complexity in terms of community height, influenced by spatial resolution.

NVDI can saturate in areas with very dense vegetation, i.e., it becomes less sensitive to additional increases in biomass or leaf cover, which also explains negative correlations between NDVI and diversity and richness. This can result in a low NDVI SD in highly vegetated and diverse areas, where variations in biomass are not effectively captured [116]. However, this was not the case in the present study. According to studies conducted by [117], several issues with NDVI have been well identified, such as its insensitivity to densely vegetated areas and its oversensitivity to changes in soil brightness due to rainfall and snowfall. Therefore, NDVI is the least appropriate choice for analyzing vegetation variation in areas with dense canopies.

In complex forests with a diverse vertical structure, such as the Atlantic Forest, NDVI may not fully capture variations in species composition. Reflectance uniformity can occur due to the presence of multiple layers of vegetation, resulting in a lower SD [118]. The negative correlation between NDVI SD and diversity and richness may also be influenced by additional environmental variables such as topography, soil, and microclimate, which affect both NDVI and species distribution, but in complex ways that may not be directly captured by NDVI SD [7].

Indices such as NDVI are sensitive to changes in chlorophyll concentration or affected by saturation at high levels of leaf area index (LAI) [119]. Canopy reflectance, in both the visible and near-infrared bands, is strongly dependent on structural properties (e.g., LAI) and biochemical properties (e.g., chlorophyll), which have similar effects on canopy reflectance, particularly in the spectral region from the green edge (550 nm) to the red edge (750 nm), posing a significant challenge in the use of these indices [119].

Potential reasons for varying results in the SVH, including effects originating from plot size in the field, were listed by [9] and [8]. The trend is that spectral variability increases as the size of the field mapping unit increases. This correlation has been demonstrated

by [27,120]. Similarly, [22] reported that plot shape can influence the results. Unlike the square plots used in the present study (30 × 30 m), elongated plots capture a broader range of ecological gradients, which may primarily affect data on gamma diversity. This could also explain the negative or null correlation values of NDVI SD found in this research and suggests that the plots used encompassed an upper canopy with dominant trees, which may have contributed to a reduced NDVI in the more advanced stages of forest succession, where higher diversity values were found.

Our results demonstrated a negative correlation between NDVI and taxonomic diversity indices. Using a multispectral RPA, previous research reported correlations ranging from weak to moderate, with $R^2$ values between 0.32 and 0.45 for NDVI applied to species diversity [16]. These values also highlight the limitations of NDVI sensitivity in certain environments, even with a spatial resolution as fine as 3 cm.

Better NDVI performance was observed with Sentinel-2 imagery at a resolution of 10 m, achieving $R^2$ values of up to 0.70 for the Shannon index in alpine forests [104]. This contrast underscores the influence of habitat type, where less dense areas, such as temperate forests or savannas, enable more effective spectral variation capture using NDVI. In our study, the high vegetation density and shadow effects likely masked spectral signals, reducing NDVI's effectiveness as a diversity predictor.

In homogeneous areas, NDVI at a 20 m resolution explained between 53% and 84% of the variance in diversity, emphasizing the importance of vegetation cover type and structural uniformity in its performance [17]. Altitudinal gradient studies employing NDVI at resolutions ranging from 2.4 to 30 m reported $R^2$ values of up to 0.87 for mean NDVI correlated with species richness [26]. This stronger relationship likely reflects less complex vertical vegetation layers and the absence of significant shadow effects, which posed substantial limitations in our study.

The analysis of different habitats using Landsat-8 images with a spatial resolution of 30 m at a landscape scale explained a moderate portion of variance in species richness, with $R^2$ values ranging from 71.7% to 75.3% [12]. However, the role of NDVI was more limited, while continuous metrics such as sdNDVI and Rao's Q better captured plant diversity in shrubland and herbaceous habitats, which are less structurally complex compared to forest ecosystems.

In grasslands, NDVI showed a positive correlation with species diversity, achieving $R^2$ values of up to 0.82 when analyzed using aerial spectroradiometry with a spatial resolution of 1 m [78]. The lower vertical complexity and absence of shadows in these habitats contributed to the higher correlation values, contrasting with the limiting factors observed in our study in dense ecosystems.

In secondary forests in northeastern China, the integration of Sentinel-2 data with LiDAR resulted in a moderate correlation between NDVI and taxonomic diversity, including Shannon's index, with $R^2$ values of up to 0.44 [121]. Although NDVI proved useful, the authors emphasized that its effectiveness in structurally complex habitats is significantly enhanced when combined with structural metrics derived from LiDAR.

*4.5. Limitations, Gaps, and Implications of the Study*

This study represents a significant contribution to the application of the Spectral Variation Hypothesis (SVH) in the Atlantic Forest, being the first to explore this relationship in one of the most biodiverse and ecologically significant ecosystems on the planet. As such, it establishes itself as an important reference for the use of high-resolution hyperspectral sensors in assessing species diversity in this tropical ecosystem. However, like any study, it presents limitations, particularly regarding the generalization of results to other ecosystems. While the findings in the Atlantic Forest are promising, the unique vegetation and biome structure make it challenging to directly apply the SVH to ecosystems with different characteristics, such as areas with lower diversity or less complex structures.

Spatial resolution was a determining factor in the observed relationship between spectral diversity and species diversity. Pixels of 1 m, 5 m, and 10 m were tested, and it

was found that the results were significantly better with a resolution of 1 m. This aligns with findings by [11], suggesting that higher resolutions enhance spectral discrimination capabilities, allowing for the capture of finer variations between species, which improves analysis accuracy. However, higher resolutions can also introduce intraspecific variability, complicating data interpretation [90]. Although 1 m pixels provided the best results, larger pixels (5 m and 10 m) also allowed spectral variations to be explored, albeit with reduced sensitivity for discriminating subtle differences in diversity.

Regarding the size of sampling plots, 9000 m² plots were used, which captured a substantial portion of ecological variability. However, as observed by [22,27,120], larger plots tend to capture more spectral variability, potentially improving the accuracy of the relationship between spectral diversity and species diversity. Although the 9000 m² plots were effective, increasing their size could have provided a more robust assessment of diversity, better representing the ecological dynamics of the ecosystem. Larger plots would help increase spatial representativeness and reduce the risk of undersampling ecological variations that could be captured at larger scales. Therefore, increasing plot size is an important consideration for future research, enabling a more comprehensive evaluation of spectral and taxonomic diversity.

In addition to plot size, the limited number of plots (n = 20) may have represented another limitation in our study. The number of plots can influence the capture of ecological heterogeneity, especially in highly diverse ecosystems such as the Atlantic Forest. A reduced number of plots may lead to underrepresentation of ecological gradients and specific habitats, while increasing the likelihood of statistical biases, such as lower statistical power and greater influence of local conditions on the analyses. However, we believe that the effects of a relatively small number of plots may have been mitigated by the fact that our plots were installed in forests of different ages, effectively sampling distinct diversity gradients. Although our results showed significant correlations between spectral and taxonomic diversity even with 20 plots, we recognize that a larger number of plots could improve the representativeness of ecological patterns captured and reduce potential biases. Future studies with an increased number of plots could more broadly explore ecological gradients and provide greater robustness to the analyses.

Another limiting factor to consider is seasonality. Phenological changes in plants affect canopy reflectance and, consequently, spectral diversity. As indicated by [26], seasonal variation can influence the relationship between NDVI and species richness, with different patterns for annual and perennial plants. Plant phenology can alter leaf cover and the presence of ephemeral plants, modifying reflectance throughout the year. In our study, the absence of multitemporal data limited a full analysis of this temporal variability, which constitutes an important limitation. However, we believe uncertainties related to seasonality are minimal since field data focused on structural attributes (e.g., tree height and diameter), which remain constant across seasons. Additionally, data collection was conducted at the end of the rainy season, when trees maintain abundant and vigorous foliage, minimizing the effects of seasonality [90,122]. Nonetheless, we acknowledge that multitemporal data in future studies would help capture seasonal variations and their effects on the relationship between spectral and taxonomic diversity.

Another critical factor to consider is understory vegetation, which can significantly influence observed reflectance. Species in the lower canopy layers may be less visible to sensors, especially in dense forests, affecting the measurement of spectral diversity. As suggested by [115], including understory data can enhance analyses, as this vegetation contributes to the structural complexity of the ecosystem. The use of LiDAR integrated with optical sensors could be a promising solution [84], as it allows capturing the vertical stratification of vegetation, improving the detection of understory species. This approach has the potential to provide more precise diversity measurements in complex tropical ecosystems such as the Atlantic Forest.

In summary, while this study provides an important foundation for the application of the SVH in the Atlantic Forest, it also highlights several limitations and gaps in the current applications of the hypothesis. Spatial resolution, the size and number of sampling plots, seasonality, and understory vegetation are factors that affect the accuracy of spectral measurements and the observed relationship with species diversity. Future research should consider multitemporal data, explore the impact of understory vegetation, test the effect of different plot sizes and numbers, and use high-resolution data to improve the accuracy of SVH and expand its applicability in complex tropical ecosystems. Additionally, the fusion of hyperspectral and LiDAR data could be a promising strategy to overcome the limitations imposed by vegetation and sensor characteristics. These advancements could not only enhance the understanding of the relationships between spectral and taxonomic diversity but also improve conservation strategies in tropical biomes and other natural ecosystems. Practically, this study demonstrates how SVH can be used to develop more accurate predictive models of species diversity, with potential applications in environmental monitoring programs and conservation management policies.

## 5. Conclusions

The Spectral Variation Hypothesis was confirmed for the Atlantic Forest biome, showing high $R^2$ values for species richness and Shannon and Simpson diversity indices calculated from illuminated 1.0 m pixel coverage. The dispersion measures (coefficient of variation—CV and standard deviation—SD) of hyperspectral bands had strong correlations with taxonomic diversity and richness, particularly in the red-edge and NIR regions of the electromagnetic spectrum.

Pixel size significantly influenced $R^2$ values, with higher values for 1.0 m pixels and lower for 10.0 m pixels. Illuminated pixels had higher $R^2$ values than those under the effects of shadows.

The use of hyperspectral RPA sensors enhances the richness of information and demonstrates potential in biodiversity studies. The key dispersion metrics selected for regression models were the mean CV, the CV of the 726.7 nm band, and the SD of the 742.3 nm band, in the red-edge region of the electromagnetic spectrum. The correlation of NDVI dispersion variables with richness and diversity was negative or null.

Spectral diversity can serve as a proxy for species diversity in the Atlantic Forest. However, factors that may affect this relationship, such as the taxonomic and spectral diversity metrics used, pixel size, and shadow effects in the images should be considered. Spectral diversity can provide estimates of species hotspots and predict spatial biodiversity patterns.

Considering the use of hyperspectral sensors in RPAs, the CV metrics of the 715.6 and 726.7 nm bands and the SD of the 742.3 nm band, correlated with species richness and the Shannon diversity index calculated from 1.0 m pixel coverage without shadow effects, can be used to estimate taxonomic diversity in the Brazilian Atlantic Forest over larger spatial extents.

Future studies should address factors not explored in this research, such as the size and number of sampling plots, seasonality, and the influence of understory vegetation. Additionally, the inclusion of sensors with varying spatial and spectral resolutions, as well as the integration of technologies like LiDAR, is recommended. These approaches have the potential to expand the applicability of the SVH, enhance conservation strategies, and strengthen environmental monitoring policies.

**Supplementary Materials:** The following supporting information can be downloaded at: https://www.mdpi.com/article/10.3390/rs16234363/s1, Table S1: Criteria for classification of succession stages of the studied areas according to Resolution No. 29/1994 of the Brazilian National Environment Council (Conama).

**Author Contributions:** Conceptualization, T.B.M.P., A.R.d.M., G.F.d.S., E.M.E. and A.Q.d.A.; methodology, T.B.M.P., A.R.d.M., G.F.d.S., E.M.E., N.M.M.R., M.M.F. and A.Q.d.A.; software, T.B.M.P., N.M.M.R. and A.Q.d.A.; validation, T.B.M.P. and A.Q.d.A.; formal analysis, T.B.M.P., A.R.d.M. and A.Q.d.A.; investigation, T.B.M.P. and A.Q.d.A.; resources, E.M.E.; data curation, T.B.M.P. and A.Q.d.A.; writing—original draft preparation, T.B.M.P. and A.Q.d.A.; writing—review and editing, T.B.M.P., A.R.d.M., G.F.d.S., E.M.E., N.M.M.R., M.M.F., J.B.B.S., C.T.d.A., H.M.D., F.G.G. and A.Q.d.A.; visualization, T.B.M.P. and A.Q.d.A.; supervision, A.R.d.M., G.F.d.S. and A.Q.d.A.; project administration, E.M.E.; funding acquisition, E.M.E. All authors have read and agreed to the published version of the manuscript.

**Funding:** This research was funded by the Espírito Santo Research and Innovation Support Foundation (FAPES)/ INOVAGRO by State Secretariat for Agriculture, Supply, Aquaculture and Fisheries (Seag) through the project "Proposing a protocol for identifying successional stages using remote sensing tools", process 2020-4KK4L.

**Data Availability Statement:** The original contributions presented in the study are included in the article/Supplementary Materials, further inquiries can be directed to the corresponding author.

**Acknowledgments:** The authors thank the Secretary of State for Agriculture, Supply, Aquaculture, and Fisheries of Espírito Santo (Seag) and the Foundation for Research Support and Innovation of Espírito Santo (FAPES) for financial support (Process 2020-4Kk4L); the Institute for Agricultural and Forestry Defense of Espírito Santo and the Laboratory of Forest Measurement and Management of the Federal University of Espírito Santo for their support in data analysis; and the Chico Mendes Institute for Biodiversity Conservation (ICMBio) for allowing the research to be conducted in the Pacotuba National Forest. JS is supported by PQ-2 Grant from Conselho Nacional de Desenvolvimento Científico e Tecnológico (CNPq). The authors would like to thank CNPq for the research grant from the author André Q. Almeida of the project "Models for estimating biomass and other dendrometric characteristics of a secondary forest" process 310299/2019-5 and "TREEcarbon: Remote forest carbon monitoring system for the Atlantic Forest and Caatinga Biomes" process 300234/2022-8. HMD is supported by Bolsa Pesquisador Capixaba (Term of Grant n° 580/2023 and Process n° 2023-5671F) from Fundação de Amparo à Pesquisa e Inovação do Espírito Santo (FAPES).

**Conflicts of Interest:** Author Fabio Guimarães Gonçalves was employed by the company Canopy Remote Sensing Solutions. The remaining authors declare that the research was conducted in the absence of any commercial or financial relationships that could be construed as a potential conflict of interest.

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
