# Peer review of "Biodiversity from the Sky: Testing the Spectral Variation Hypothesis in the Brazilian Atlantic Forest"

_remotesensing, doi:10.3390/rs16234363_

Round 1

Reviewer 1 Report

Comments and Suggestions for Authors

Review of the Manuscript "Biodiversity from the Sky: Testing the Spectral Variation Hypothesis in the Brazilian Atlantic Forest"

The study explores the relationship between species diversity and hyperspectral data collected by a remotely piloted aircraft in secondary tropical forest areas within the Brazilian Atlantic Forest biome. The authors aim to test the Spectral Variation Hypothesis (SVH), which posits that spectral diversity, measured using metrics like the coefficient of variation (CV) and standard deviation (SD) of hyperspectral bands, correlates with species richness and diversity. The study examines how pixel size and shadow effects influence this relationship and confirms the hypothesis for this biome, with strong correlations particularly in the red edge and near-infrared regions of the electromagnetic spectrum.

The manuscript presents interesting and valuable research on biodiversity monitoring using hyperspectral remote sensing in a biodiverse region. However, there are several areas that need improvement before it can be accepted. The authors should address methodological concerns, clarify the novelty and significance of their approach compared to previous studies, and improve data interpretation to make the conclusions more robust.

Major Comments:

1. The introduction presents the context of biodiversity monitoring well, but the novelty of the research could be more clearly articulated. What specific gaps in the current literature does this study address, and how does it advance knowledge in this area?

2. The methods section is detailed but lacks justification for some of the choices, such as why certain pixel sizes (1.0, 5.0, and 10.0 m) were selected for analysis. More explanation is needed on how shadow effects were addressed and why NDVI was not selected as a primary metric.

3. While the study finds strong correlations between spectral diversity and species richness/diversity, the explanation for why the red edge and NIR regions perform better should be further elaborated. Additionally, the authors should discuss the limitations of these findings, especially regarding the generalizability of the results to other ecosystems.

4. The statistical tests used for variable selection and model validation are appropriate, but more details are needed about the robustness of these models. Were any alternative models considered? What are the potential biases in the data, especially given the small number of sample plots (n = 20)?

Minor Comments:

Lines 44-61: The introduction mentions the importance of studying tropical forests and species diversity. However, the discussion lacks a critical evaluation of the gaps in current SVH applications. Consider expanding this section to articulate the specific challenges that this study addresses, such as the impact of shadow effects and pixel size on SVH application.

Lines 87-89: The question raised about NDVI as the best vegetation index (VI) is relevant, but the introduction could benefit from a more structured explanation of why hyperspectral data offer advantages over traditional multispectral data for this study.

Line 142–146: In the description of the study area, clarify the criteria used to classify the forest into early, intermediate, and advanced successional stages. It is not clear whether these classifications are based purely on age or also on structural or ecological characteristics.

Lines 223-237: The process for selecting hyperspectral metrics through the leaps package and BIC values could be clarified. Including a flowchart or diagram explaining the decision-making process for selecting metrics (e.g., SD, CV) would aid reader comprehension.

Lines 240-266: The results mention significant changes in correlation across the spectrum depending on pixel size and shadow effects. More attention could be given to explaining the ecological significance of these changes. What do the observed trends reveal about species diversity in the Atlantic Forest? Why might shadow effects have a stronger impact on species richness than on Shannon and Simpson indices?

Lines 280-291: The selection of specific metrics (e.g., SD 742.3 nm for richness) is interesting, but more explanation is needed about why these particular wavelengths are most informative. Additionally, I recommend further elaboration on the ecological implications of using these bands, particularly in terms of their sensitivity to forest composition and structure.

Line 299–307: The effect of shadow on species detection is discussed, but the methodology for controlling or mitigating shadow effects is unclear. Was shadow masking performed, or were pixels affected by shadow included in the analysis?

Reviewer 2 Report

Comments and Suggestions for Authors

This study was to test the Spectral Variation Hypothesis through associations between species diversity and richness measured in the field and hyperspectral data collected by a Remotely Piloted Aircraft (RPA) in areas with secondary tropical forest in the Brazilian Atlantic Forest biome. The writing is good and the entire structure is smooth and logic. Below are some minor comments.

It is better to summarize the research methods for spectral diversity and provide a table of key literature and methods in the introduction section.

Line 71: it is better not to start a sentence with a reference number.

What errors and uncertainties might arise from the field survey time and remote sensing image time being in different seasons?

Would the spectral characteristics of mixed pixel decomposition be better than the variables calculated using the original spectra? The authors can have a test.

Figure 3 needs to be polish.

Line 580 do not start a sentence with reference numbers.

The author should compare and analyze the results of this study with previous research.

The discussion section should include the broader implications of this study.

Round 2

Reviewer 1 Report

Comments and Suggestions for Authors

The author’s responses are thorough and demonstrate an effort to address each of my comments. They have made significant revisions to the manuscript, including adding new sections, figures, and clarifications.